# Association between Noise Annoyance and Mental Health Outcomes: A Systematic Review and Meta-Analysis

**DOI:** 10.3390/ijerph19052696

**Published:** 2022-02-25

**Authors:** Xiangpu Gong, Benjamin Fenech, Claire Blackmore, Yingxin Chen, Georgia Rodgers, John Gulliver, Anna L. Hansell

**Affiliations:** 1Centre for Environmental Health and Sustainability, University of Leicester, Leicester LE1 7HA, UK; xg82@leicester.ac.uk (X.G.); clb91@leicester.ac.uk (C.B.); yc310@leicester.ac.uk (Y.C.); jg435@leicester.ac.uk (J.G.); 2National Institute for Health Protection Research Unit in Environmental Exposures and Health, University of Leicester, Leicester LE1 7HA, UK; 3Noise and Public Health Group, Environmental Hazards and Emergencies Department, UK Health Security Agency, Birmingham B2 4BH, UK; benjamin.fenech@phe.gov.uk (B.F.); georgia.rodgers@phe.gov.uk (G.R.)

**Keywords:** environmental and neighborhood noise, traffic noise, noise annoyance, mental health, depression, anxiety disorder, general mental health

## Abstract

To date, most studies of noise and mental health have focused on noise exposure rather than noise annoyance. The purpose of this systematic review and meta-analysis was to evaluate whether the available evidence supports an adverse association between noise annoyance and mental health problems in people. We carried out a literature search of Web of Science, PubMed, Scopus, PsycINFO, and conference proceedings published between 2000 and 2022. Thirteen papers met the inclusion criteria. We conducted meta-analyses of noise annoyance in relation to depression, anxiety, and general mental health. In the meta-analyses, we found that depression was approximately 1.23 times greater in those who were highly noise-annoyed (N = 8 studies). We found an approximately 55% higher risk of anxiety (N = 6) in highly noise-annoyed people. For general mental health (N = 5), highly annoyed participants had an almost 119% increased risk of mental health problems as assessed by Short Form (SF) or General Household Questionnaires (GHQ), but with high heterogeneity and risk of publication bias. In conclusion, findings are suggestive of a potential link between noise annoyance and poorer mental health based on a small number of studies. More evidence is needed to confirm these findings.

## 1. Introduction

Mental and addictive disorders were estimated to affect over 1 billion people worldwide in 2016, accounting for 7% of the global burden of disease as measured in disability-adjusted life year (DALYs) and 19% of all years lived with disability [1]. Mechanistic and epidemiological evidence suggests that exposure to traffic noise could be associated with poorer mental health in the population, either directly or via noise annoyance. Noise annoyance is a stress reaction to environmental noise [2], which is thought to be linked to the release of catecholamines from the hypothalamic–pituitary–adrenal axis [3]. Repeated noise annoyance may increase the risk of higher stress-hormone exposures [3], which could be associated with a variety of mental health disorders [4].

The link between noise exposure and mental health disorders is garnering increasing attention because noise pollution has long been a persistent urban problem in developed countries. Environmental Noise Guidelines for the European Region recommend reducing road and railway traffic noise levels to under 53 and 54 decibels (dB) Lden, respectively, as noise levels exceeding these thresholds have been linked to adverse health effects [5]. However, the European Commission estimated that 125 million people in Europe are exposed to noise levels greater than 55 decibels Lden from road traffic, with over 37 million exposed to noise levels greater than 65 dB Lden [6]. England’s third round of noise mapping, conducted in 2017, found that approximately 11.54 million and 1.50 million people living inside and outside agglomerations (major urban areas), respectively, were exposed to noise levels greater than 55 decibels Lden from roads and railways [7,8,9].

Systematic literature reviews to date have found statistically significant associations between aircraft noise exposure and depression in the general population, but not between noise from other sources and other mental health outcomes [2,10,11,12,13]. A number of studies, however, have reported statistically significant associations between noise *annoyance* and mental health outcomes for neighbourhood [14], road traffic [14], and aircraft noise [14,15]. The complex relationship between noise and mental health, including the mediating effect of noise annoyance, remains an under-researched area, but could provide mechanistic insight into the link between noise exposure and mental health issues [16].

The purpose of this systematic review and meta-analysis is to examine whether existing studies support a negative association between high noise annoyance and mental health outcomes in people. Highly annoyed participants are defined as individuals who in a questionnaire selected “very” or “extreme” on a 5-point verbal scale for annoyance ($HA_{V}$), the top three highest values on an 11-point numeric scale ($HA_{N}$), or the weighted top two verbal responses for the 5-point verbal question ($HA_{VW}$), as recommended by ISO/TS 15666:2021 [17]. 

The mental health outcomes of concern include depression and anxiety disorder, which affect approximately 4.4% and 3.6% of the global population, respectively [18]. We also investigated the relationship between high noise annoyance and general mental health.

## 2. Methods

This systematic review and meta-analysis aims to examine whether high noise annoyance can be associated with negative mental health outcomes. We conducted the study in accordance with PRISMA guidelines [19,20], and synthesised evidence using the PECCOS (population, exposure, comparator, confounder, outcome, and study design) procedures used for the systematic reviews underpinning the WHO 2018 Noise Guidelines for the European Region [5,21,22].

In our systematic review and meta-analysis, three reviewers (X.G., C.B., Y.C.) independently selected relevant papers identified through a comprehensive literature search and extracted data using a standardised proforma. 

### 2.1. Paper Identification

We identified papers through searches of four databases, manual searches of relevant conference proceedings, referrals from colleagues, and review of papers identified in systematic reviews examining the mental health effects of noise exposure [2,11,12,23,24,25]. see Appendix A Table A1 shows the full list of conferences and search terms used to scan proceedings.

We (X.G., Y.C., B.C.) searched the Web of Science, PubMed, Scopus, and PsycINFO databases from 2000 to January 2022 for studies that examined the relationship between annoyance from any noise sources and the mental health outcomes of interest. See Appendix A Table A2 contains the search terms used in Web of Science, PubMed, Scopus, and PsycINFO.

The search results were imported into EndNote. After eliminating duplicates, XG, CB, and YC independently screened the remaining studies using the PECCOS inclusion and exclusion criteria listed in Table 1 [5]. Since the purpose of our research is to quantify the relationship between high noise annoyance and mental health in people, we excluded any papers that could not be included in a quantitative meta-analysis. Disagreements were resolved through discussion. 

### 2.2. Definition of Outcomes

We primarily focused on anxiety and depressive disorders. However, a significant proportion of published research examines people’s overall mental health, which may be associated with but not classified as depression or anxiety disorders. We therefore also looked at general mental health as a third outcome category.

Most relevant studies on depression and/or anxiety disorders relied on either self-reported disease diagnoses (SRD) or self-reported use of psychotropic medications (SRM), such as antidepressants and anxiolytics. In one study, validated questionnaires (VQ) such as the Patient Health Questionnaire-9 (PHQ-9; for depression) and the Generalized Anxiety Disorder-2 (GAD-7; for anxiety disorder) were used to detect cases of depression or anxiety disorders by comparing participant scores to cut-off values. One study identified cases using all three of the methods outlined above: SRD, SRM, and VQ. Another study screened for depressive and/or anxiety symptoms using unvalidated questionnaires (UQ).

Relevant publications on general mental health used a variety of instruments that can be classified into two broad categories. The first group comprises two versions of the General Health Questionnaire: GHQ-12 and GHQ-30. We refer to GHQ-12 and GHQ-30 collectively as GHQ studies. The second includes the Short Form Survey; there are multiple versions of Short Form surveys commonly used in relevant studies. They include SF-36 (and its derivative MIH-5) and SF-12 (a shorter version of SF-36). We refer to SF-12, SF-36, and MIH-5 collectively as SF studies. These screening tools for general mental health have varying scales, but we included only studies that used cut-off values to dichotomise outcomes as cases or non-cases.

We did not consider perceived stress levels as there was only one study that examined this outcome using the Perceived Stress Scale (PSS). 

All of the outcomes are binary, which allows for statistical comparisons of the estimates.

### 2.3. Definition of Exposure

We restricted our analysis to annoyance caused by any sources of environmental and neighbourhood noise.

The 11-point numeric noise-annoyance scale (range 0–10; a higher number indicates a greater degree of annoyance) and the verbal 5-point response scale (1 “Not at all”, 2 “Slightly”, 3 “Moderately”, 4 “Very”, and 5 “Extremely”) are two frequently used questionnaires for identifying noise annoyance.

We adopted three definitions of high noise annoyance in accordance with ISO/TS 15666:2021 [17]. The first is $HA_{N}$, which uses the top 3 points (8, 9, and 10) of the 11-point numeric noise-annoyance scale to identify highly annoyed participants [2,17,26]. $HA_{V}$ uses the upper two steps (4 “Very” and 5 “Extremely”) of the verbal 5-point response scale to define highly annoyed individuals [2,17,26,27,28].

Because the $HA_{V}$ method’s cut-off value of 60% is lower than the 72% employed in research using the $HA_{N}$ approach [28], a third definition, $HA_{VW}$, was proposed, which uses the same 5-point verbal scale but weights “Very” by 0.4 and “Extremely” in full to produce a mathematical similarity between the former two approaches [17].

$HA_{VW}$ has a mathematical cut-off value that is similar to $HA_{N}$. Although $HA_{V}$ has a lower cut-off threshold than $HA_{N}$, the verbal questions may be interpreted differently from the numerical questions [17]. They both detect levels of annoyance that are not considered trivial or moderate [28]. 

Most relevant studies used either $HA_{V}$ or $HA_{N}$ to identify highly annoyed participants, but none used $HA_{VW}$. 

Additionally, there were studies that used 5- or 11-point scales but only made mention of being “disturbed” or “bothered” by noise on the questionnaires. We treated these questionnaires comparable to standard ones, considering being disturbed or bothered as elements of annoyance [2]. This allowed us to include two additional studies into the meta-analysis.

To increase the number of studies included in the meta-analyses, we also included studies that employed a three-point scale. We chose the highest score as indicative of high annoyance.

We considered perception of noise to be fundamentally different from the three components of noise annoyance as defined by Guski [2] (disturbance, emotional and cognitive response). Therefore, we excluded publications that used noise perception as the exposure variable.

$HA_{N}$, $HA_{V}$, as well as other variables of high annoyance are binary, with one value indicating highly annoyed and the other otherwise.

### 2.4. Effect Size Extraction

We combined all of the studies for each outcome, regardless of the source of the noise, on the assumption that the annoyance was having the same biological effect on people. 

We used odds ratios as the unified effect size, because all studies but one used logistic regressions to analyse data and reported odds ratio. Eze [29] reported relative risk. We converted the relative risk and 95% CI into odds ratio by using the formula $OR\approx RR^{2}$, assuming that mental health is a common health problem among participants (reported by >15% participants) [30].

We extracted estimates whenever possible from models in which noise annoyance as the only noise exposure variable. Two studies [15,31] (both focused on general mental health) presented results from models that incorporated both noise levels and noise annoyance, with noise annoyance potentially serving as both an exposure and a mediator. Given the low number of studies available in total, we also included these in meta-analysis; sensitivity analyses excluding these papers did not lead to effective changes in results or interpretation. 

We derived the estimates from the fully adjusted model for each paper. If there were multiple estimates from the fully adjusted specification, the most conservative (lowest in size) coefficients were then extracted. For instance, Schreckenberg [31] provided two estimates of the relationship between noise annoyance and general mental health, based on the SF-12 and SF-36 mental health scales, respectively [31], and we selected the SF-12 estimate. Eze [29] reported findings using both a full sample and a sample of non-movers, the latter of which was used. 

### 2.5. Risk of Bias

Bias risk was assessed using the checklist in see Appendix A Table A3 of the Methodology for Systematic Evidence Reviews for WHO Environmental Noise Guidelines for the European Region [19]. The checklist contains five domains and a total risk of bias. For each study, the total risk is considered low when at least 4/5 domains are judged to be of low risk of bias, including domains 1, 2, and 3. Any study that does not meet this criterion is deemed high risk. Please see the Methodology document [19] for full details.

We created figures to summarise the risk of bias using the R package *robvis* [32].

### 2.6. Statistical Analysis

We estimated pooled odds ratio and 95% confidence intervals (CI) using random-effects meta-analysis. The random-effects meta-analysis relies on an assumption that the exposure effect from each individual study might be different [33,34], which enables the regression to incorporate sources of heterogeneity [33]. The analysis was carried out using the *metan* package [35] in Stata 17 [36]. We log-scaled odds ratios and 95% confidence intervals to make data nearly symmetrical for the meta-analysis. We reported exponentiated pooled effects and 95% CI. 

We examined the pooled association between annoyance caused by any types of noise and mental health problems. Due to insufficient studies, we were unable to analyse the relationships relating to noise annoyance from specific sources, e.g., traffic or neighbourhood.

We hypothesised that pooled analyses of depression or generalised anxiety disorder determined by either self-reported diagnosis or questionnaire (SRD/VQ/UQ) or self-reported medication (SRM) may exhibit high heterogeneity due to studies detecting varying degrees of severity. Studies that used SRD/VQ/UQ to screen for depression and anxiety disorder may identify patients with a broader spectrum of severity. By contrast, individuals suffering from moderate-to-severe mental health problems were likely to be included in studies that used SRM to identify cases. Thus, we conducted subgroup analyses by dividing studies into SRD/VQ/UQ and SRM. The study that identified cases using SRD, VQ, and SRM was assigned to the SRM subgroup.

Moreover, studies on general mental health used two broad categories of validated instruments: the GHQ and the SF. These two instrument families appear to assess different aspects of mental health [37], that may introduce heterogeneity into the meta-analysis. We thus performed meta-analysis on subgroups and divided samples into GHQ and SF.

To assess the effect of outliers on our findings, we used leave-one-out analysis to recalculate the pooled effects multiple times by omitting one study from each analysis.

To assess publication bias, funnel plots were used. Each plot depicted the effect size of each study on the X axis and the standard error on the Y axis.

### 2.7. Quality of Evidence

The overall quality of evidence was judged according to the adapted version of the Grading of Recommendations Assessment, Development, and Evaluation (GRADE) guidelines, as used in systematic reviews of noise and health conducted to develop the 2018 WHO Noise Guidelines for the European Region [22,38].

## 3. Results

We found 350 articles in Web of Science, PubMed, Scopus, and PsycINFO database searches. One additional record was identified through reviewing conference proceedings and literature reviews. We removed 105 duplicates and additional 190 articles that did not meet the inclusion criteria after screening titles and abstracts. Following a full-text analysis, we eliminated 42 papers for the reasons listed in Figure 1. This left us with 13 papers for review (listed in Table A4). The full description of studies is presented in Table A6.

The average number of participants was around 7427 (range 1244 to 19,294). The participants were selected from the general population (N = 6), the male population (N = 1), the general population living near airports (N = 4), and the general population living in multistorey houses (N = 2). All studies were conducted in European countries. Except for two longitudinal studies [39,40], the studies were all cross-sectional.

One study assessed noise annoyance using both 5-point and 11-point scales. Others used either 3-point (N = 2), 5-point (N = 7), or 11-point (N = 3) scales. Two studies used $HA_{N}$ while two used $HA_{V}$ to define high annoyance. The remaining studies did not use standard definitions of high annoyance as suggested by Clark [17]. The sources of noise annoyance included aircraft (N = 8), road vehicles (N = 6), trains (N = 3), neighbourhood (N = 6), industrial (N = 2), and unspecific traffic sources (N = 1).

Seven studies examined both depression and anxiety disorders [40,41,42,43,44,45], while two focused exclusively on depression [29,46]. Measures used in these studies included the intake of antidepressants (N = 4) and anxiolytics (N = 3), as well as self-reported physician diagnosis of depression (N = 2). Beutel [40] and Beutel [43] used PHQ-9 and GAD-2 to screen for depression and anxiety disorder. Jensen [47] used unvalidated questionnaires to identify the case of depression and anxiety.

Five papers used self-reported mental health measures. Baudin [15] used GHQ-12 and defined cases as those with a score ≥ 3 on the scale. Stansfeld [48] adopted a threshold of 4 on GHQ-30. Schreckenberg [31] and Jensen [41] used the SF-12, and the cut-off values in these two studies were median and 52 on the scale, respectively. MHI-5 was used by Hammersen [14] with a cut-off value of 52.

### 3.1. Risk of Bias

Figure A1 and Figure A2 illustrate the detailed evaluations of each paper against each criterion. More than three-quarters of studies had a high risk of bias. Two primary reasons for this were domain 1—a lack of standardised definitions of high noise annoyance being used (8 studies); and domain 3—a study response rate below 60% (N = 3). An additional reason for high bias risk ratings was blinding (N = 3). Finally, using unvalidated mental health questionnaires contributed to a high risk of bias score for three studies.

### 3.2. Meta-Analysis Results

#### 3.2.1. Depression

There were eight studies available, of which six were included in the meta-analysis, as two studies used the same dataset; we selected Baudin [45] (using data from HYENA and DEBATs studies) over Floud [44] (using data from HYENA only) and Beutel [40] (using data from Gutenberg Health Study—longitudinal design) over Beutel [43] (using data from Gutenberg Health Study—cross-sectional design).

As illustrated in Figure 2, the pooled odds ratio for the forest plot for all six studies was 1.23 (95% CI [1.03, 1.48]). However, $I^{2}$ and Q were 60.4% and 12.63, respectively, implying significant heterogeneity between studies.

One potential source for the high degree of heterogeneity was the difference in the way in which measurement of depression was made. The pooled coefficient for three studies that used SRD or VQ was 1.50 (95% CI [1.03, 2.19]) and significant. Although $I^{2}$ and Q remain high in SRD or VQ studies, subgrouping significantly reduces the heterogeneity between studies that used SRM as the outcome. The effect for this subgroup was 1.08 (95% CI [1.01, 1.16]), which was statistically significant and with low $I^{2}$ and Q.

A leave-one-out analysis (see Appendix B Figure A3) indicates that Jensen [47] was probably an outlier, likely owing in part to the study’s use of an unvalidated questionnaire to detect depression and anxiety.

#### 3.2.2. Anxiety Disorder

We pooled data from four out of six relevant studies to assess the association between high noise annoyance and anxiety disorder. Again, we selected Baudin [45] (using data from HYENA and DEBATs studies) over Floud [44] (using data from HYENA only) for the same reason as stated previously, and Beutel [40] (using data from Gutenberg Health Study; longitudinal design) over Beutel [43] (using data from Gutenberg Health Study; cross-sectional design).

The forest plot in Figure 3 indicates that the pooled effect based on all four studies was 1.55 (95% CI [1.14, 2.10]), with large $I^{2}$ and Q, suggesting significant heterogeneity between studies.

When samples were divided into two subgroups based on outcome assessment methods (VQ/UQ vs. SRM), we again observed small $I^{2}$ and Q for the studies that used SRM to measure anxiety disorder. The odds ratio for the SRM subgroup was 1.44 (95% CI [1.15, 1.81]). Across studies that used VQ/UQ to detect anxiety disorder, the pooled association was much greater in size (OR = 1.73 95% CI [0.82, 3.66]), albeit non-significant. This subgroup has significant heterogeneity as suggested by the large $I^{2}$ and Q.

Figure A4 depicts the results of a leave-one-out analysis, which again suggests Jensen [47] was probably an outlier.

#### 3.2.3. General Mental Health

Six estimates were available from five studies (one study reported results separately for men and women), all of which used validated instruments to assess mental health that fall into two categories: GHQ and SF.

In Figure 4, the pooled effect is 2.19 (95% CI [1.49, 3.23]). However, $I^{2}$ is 94.10% and Q is 85.06, indicating a high degree of heterogeneity across studies. Subgroup analysis results show that a high level of noise annoyance was associated with an almost threefold increased risk of self-reporting a mental health problem (OR = 3.17, 95% CI [1.69, 5.95]), based on two studies that used either GHQ-30 or GHQ-12. The pooled odds ratio for the three SF studies was 2.00 (95% CI [1.27, 3.15]). According to their $I^{2}$ and Q, there appears to be significantly more heterogeneity across SF studies than across GHQ studies as judged by $I^{2}$ and Q.

We conducted a leave-one-out analysis, as shown in Figure A5, and identified Schreckenberg [31] using the SF-12 as a potential outlier.

### 3.3. Publication Bias

Funnel plots in Figure A6 (depression studies) and Figure A7 (anxiety studies) illustrate a relatively symmetric funnel shape between studies that use SRM to identify cases. However, funnel plots for other subgroups of depression and anxiety studies, as well as for general mental health (Figure A6, Figure A7 and Figure A8) indicated an asymmetric shape, suggesting a high risk of publication bias.

### 3.4. Quality of Evidence

The quality of evidence using the GRADE system is presented in Table A5. We chose to separately assess each subgroup for depression and anxiety disorder included in the meta-analysis due to the significant heterogeneity within each domain. All subgroups and general mental health began with “low” ratings, which was consistent with the cross-sectional design used in all, but two studies. We rated evidence as ‘very low’ quality for all depression and anxiety subgroups, as well as the general mental health group.

## 4. Discussion

We conducted a meta-analysis of high annoyance from environmental and neighbourhood noise and three domains of mental health problems: depression, generalised anxiety disorder, and general mental health.

Our results (N = 13) show significant pooled associations between high noise annoyance and all three domains, albeit with a high degree of heterogeneity.

In subgroup analyses, we divided relevant studies according to health outcome identification (self-reported diagnosed (SRD) or validated questionnaire (VQ) or unvalidated questionnaire (UQ) detected vs. self-reported medication intake (SRM)) for each domain of mental health problem. We found a statistically significant correlation between high noise annoyance and psychotropic medication use (antidepressant or anxiolytic) with a significantly low level of heterogeneity.

The coefficient size for anxiolytics was consistently larger than that for antidepressants, based on a small number of studies. Notably, a recent study that focused on actual noise levels rather than noise annoyance discovered a significant correlation between road noise levels and prescriptions for anxiolytics, but not for antidepressants [44]. Anxiolytics can be prescribed for sleep problems [44], which may contribute to a relationship between noise/noise annoyance and anxiolytics intake. Evidence to date, however, found a non-significant link between noise level and the prescription of hypnotics [44,48]. More detailed studies are needed to determine whether noise annoyance is related to moderate- to-severe anxiety or whether it is associated with sleep disturbance.

We combined all estimates of high noise annoyance regardless of the source of the noise. SRM-based studies on depression and anxiety disorders and GHQ-based studies on general mental health evidenced a low level of heterogeneity across studies. This supports our previous hypothesis that annoyance from environmental and neighbourhood noise may have the same biological effect irrespective of its source.

The proposed biological mechanism underlying the noise annoyance and mental health relationship is that noise exposure may induce the release of stress hormones [3,49], disrupting hormonal rhythms via activation of the Hypothalamic–Pituitary–Adrenal (HPA) axis [49]. Dysregulation of the HPA axis is significantly associated with a variety of mental health disorders, including depression, PTSD, etc. [4,50], which leads to a hypothesised link between noise exposure and mental health problems in humans. Noise annoyance is a proxy for the dissatisfaction and distress associated with noise exposure [51], implying that noise annoyance may act as a mediator between noise exposure and health outcomes [52,53,54]. This may explain why we found a strong relationship between noise annoyance and mental health, whereas other meta-analyses to date have discovered only limited evidence of the relationship between actual noise levels and mental health outcomes. A 2019 meta-analysis by Dzhambov [25] found a positive—albeit non-significant—correlation between road traffic noise levels and depression or anxiety disorder. A meta-analysis by Hegewald [12] published in 2020 also identified a non-significant increase in depression risk associated with a 10 dB increase in railway or road traffic noise, but a statistically significant higher risk of depression associated with the same increase in aircraft noise. There was an insufficient number of studies to meta-analyse the pooled relationship between noise exposure from aircraft and general anxiety disorder, as noted by Hegewald [12] and Dzhambov [25].

One issue in the interpretation of an association between noise annoyance and mental health is reverse causality. A competing theory argues that mental health may be a context factor that increases vulnerability to environmental stressors, and that noise annoyance, as a psychological response to stress, may be a result of poor mental wellbeing [55]. Evidence on causal directions is still very limited. One study used structural equation modelling (SEM) to investigate the causal direction of the relationship between aircraft sound exposure, aircraft noise annoyance, and mental-health-related quality of life (HQoL) [52]. Both annoyance and mental HQoL measured at survey wave one had an impact on mental HQoL and annoyance measured at survey wave two, suggesting that annoyance and mental HQoL are reciprocally associated with each other. The mediation effect of aircraft noise annoyance was found to be considerably higher than the mediation effect of mental HQoL, indicating that the effect of mental HQoL on annoyance is independent from sound exposure. In two of the three SEM models investigated, the direct effect of aircraft sound exposure on mental HQoL was not significant; that is, annoyance fully mediated the relationship between aircraft noise exposure and mental HQoL [52].

We cannot rule out either explanation based on the small number of studies and their cross-sectional nature. Further research is urgently needed to investigate the causal relationship between noise annoyance and mental health in people.

We focused on high annoyance from noise as the exposure because it is generally well defined and examined [2,17,26,28]. Being disturbed, bothered, and annoyed are common feelings to daily nuisances. By concentrating on individuals who exhibited a high level of annoyance due to noise exposure, we are more likely to disentangle chronic stress responses from shorter-term negative experiences [26,27]. Furthermore, it is generally accepted that high annoyance is more likely to have clinical significance [56].

A strength of our study is that to the best of our knowledge, this systematic review and meta-analysis is the first to consider associations between noise annoyance (rather than noise levels) and mental health. We identified some possible sources of heterogeneity and conducted subgroup analysis. This contributed to a reduction in the degree of heterogeneity across some subgroups. Further research should also investigate potential differences between men and women in associations between high noise annoyance and depression and anxiety.

Limitations to our study include the fact that most studies used in our meta-analysis and systematic review were cross-sectional, limiting the ability to establish causality in the association between mental health and high noise annoyance. We identified that only a small number of studies are available, with some heterogeneity in both the exposure assessment and outcome assessment, and the grading of evidence as low-quality. We were unable to consider participant age ranges in the meta-analysis. A final limitation is that we used a non-mesh search strategy, which may introduce errors that could compromise the quality and validity of our systematic review [57].

Our findings, combined with limited evidence from longitudinal analyses of epidemiological data, suggest that high noise annoyance is potentially an important mediator of the relationship between noise exposure and mental health outcomes. This is of concern if noise annoyance has increased in recent years, as suggested by some studies [2,53,58]. Interventions to reduce the burden of ill health attributable to environmental and neighbourhood noise should focus on both noise exposure and noise annoyance.

## 5. Conclusions

To the best of our knowledge, this systematic review and meta-analysis is the first to consider associations between noise annoyance (rather than noise levels) and mental health. Our results suggest a negative link between high noise annoyance and depression, generalised anxiety disorder, and general mental health, based on a small number of studies. This finding supports the hypothesis that noise annoyance may be negatively associated with mental health problems in individuals. More studies are needed to investigate this further, but these tentative associations may suggest that public health interventions should focus on reducing noise annoyance as well as noise exposure.

## Figures and Tables

**Figure 1 ijerph-19-02696-f001:**
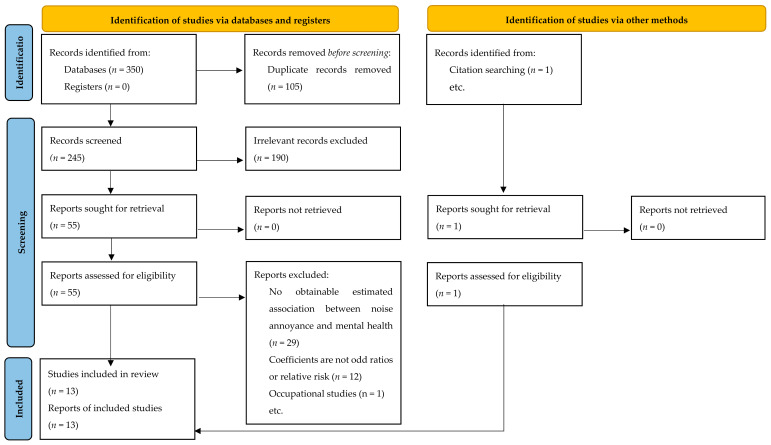
PRISMA flow chart showing number of papers identified [20].

**Figure 2 ijerph-19-02696-f002:**
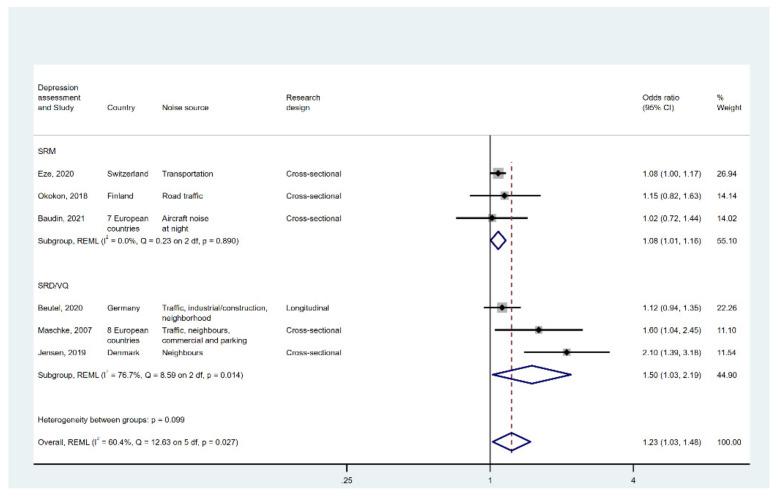
Forest plot displaying the link between high noise annoyance and depression. Note: weights and between-subgroup heterogeneity text are from random-effects model.

**Figure 3 ijerph-19-02696-f003:**
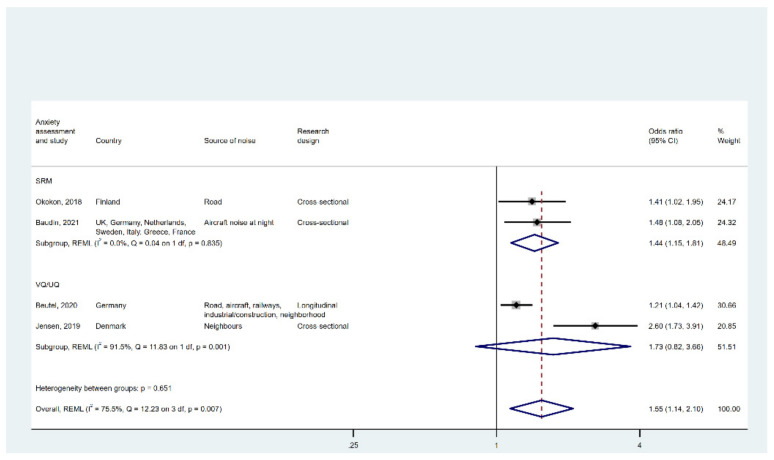
Forest plot displaying the link between high noise annoyance and anxiety disorder. Note: weights and between-subgroup heterogeneity text are from random-effects model.

**Figure 4 ijerph-19-02696-f004:**
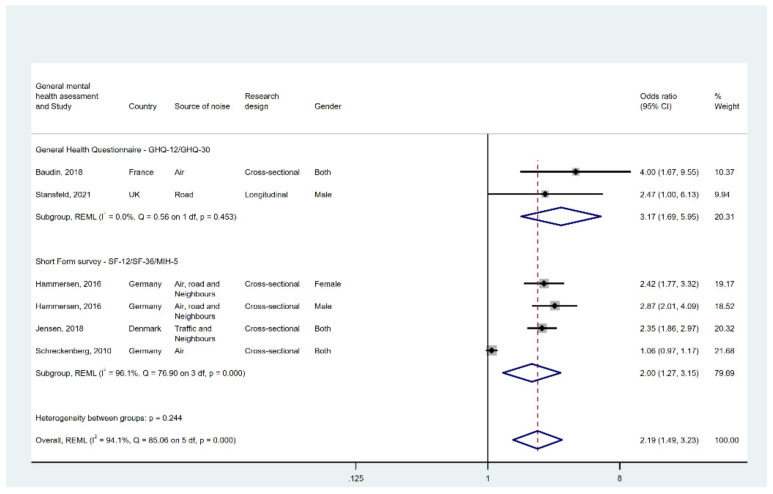
Forest plot displaying the link between high noise annoyance and general mental health. Note: weights and between-subgroup heterogeneity text are from random-effects model.

**Table 1 ijerph-19-02696-t001:** PECCOS review inclusion and exclusion criteria.

Category	Inclusion	Exclusion
Population	We considered studies that examined the general adult population, or a subgroup of the general adult population, such as men or women.	
Exposure	We restricted noise sources to environmental or neighbourhood noise from road, rail, aircraft, commercial, industrial, wind turbine, and construction activities. To assess noise annoyance, questionnaires were limited to standard annoyance questionnaires (5-point verbal question or 11-point numeric question) or questionnaires that mentioned noise disturbance or bothering.	We excluded studies examining occupational noise exposure or noise perception.
Confounders	No inclusion confounder criteria were used, following methods used for the systematic reviews underpinning the 2018 WHO Noise Guidelines for the European Region [5].	
Outcomes	We considered studies that assessed mental health outcomes using objective or self-reported measures, such as diagnosis of disease or prescription of drugs. We also included studies that implemented mental health screening tools but dichotomised the outcomes as cases or non-cases.	We excluded studies that used mental health screening tools but did not dichotomise the outcomes.
Study types	Cross-sectional Longitudinal, Prospective and retrospective cohort, Case-control, and Experimental studies with quantitative results.	

## Data Availability

All the data were obtained from previously published papers.

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
