# Peer review of "Association between Noise Annoyance and Mental Health Outcomes: A Systematic Review and Meta-Analysis"

_ijerph, 2022, doi:10.3390/ijerph19052696_

Round 1

Reviewer 1 Report

This manuscript deals with an important topic, the association between noise and mental health outcomes. The inclusion of different symptoms of mental health outcomes is a strength of this systematic review. However, there are some issues that should be addressed to enhance the paper’s contribution.

General Comments:

  1. While in some studies, noise annoyance is considered as an outcome variable in relation to actual noise level, in other studies, noise annoyance is considered as an exposure variable. Have the 12 studies that were included considered actual noise levels? What is the role of noise annoyance in the 12 studies (is it an exposure variable or a mediator)?

  1. It would be helpful to clarify why you do not include psychological stress as one of symptoms of mental health outcomes in the systematic review.

  1. The study included studies consider noise from different sources (e.g., traffic, commercial, industrial, neighborhood). However, regarding the different sources of noise, the underlying psychological mechanism of noise annoyance and its related mental health outcomes might be different. Noise is not only a physical concept, but also a socio-psychological concept. Noise from different sources might have different socio-cultural meanings. Therefore, it may further impact the relationship between noise annoyance and its related mental health outcomes differently. It can be helpful for this study to distinguish findings of the studies that have focused on the transportation noise and the neighborhood/commercial noise. Alternatively, it is helpful to merely review studies that have focused on transportation noise.

  1. This study only searched the Web of Science and PubMed databases. More databases (Scopus, Embase, PsycINFO, etc.) should be considered and included in the search stage.

Specific Comments:

  1. In Table 1, “Outcomes”, the authors mentioned that “We considered studies that assessed mental health outcomes using objective measures, such as diagnosis of disease or prescription of drugs.” But it seems in the “Definition of Outcomes”, studies also used “self-reported measures”.

  1. In the Table A.6 Full description of studies, it would be helpful to add a column to summarize whether these studies consider actual noise levels, as well as the range of the actual noise levels in these studies.

Reviewer 2 Report

Dear authors,

I enjoyed reading this review. I think it is interesting, topical and relevant. For the most part it is easy to read and understand. Although there are some aspects that need to be completed and revised.

--------------------------------------------------------

Review ijerph-1510823

Title: Association between noise annoyance and mental health outcomes – a systematic review.

Dear authors,

I enjoyed reading this review. I think it is interesting, topical and relevant. For the most part it is easy to read and understand. Although there are some aspects that need to be completed and revised:

Title

  1. It is necessary to include in the title "Systematic review and meta-analysis". It is a recommendation in item 1 of the Prisma Meta-Analysis Extension (checklist).

Abstract

  1. It is important that you include the objective of the review, or at least the research question.
  2. Line 17: It is not clear what you mean by "assuming that the annoyance was having the biological effect". Please, concrete this.
  3. Line 22: Short Form of which questionnaire? Please, clarify.
  4. To reduce words, you can put "n=3" instead of "three studies". Just a suggestion.

Introduction

  1. Line 32: Define DALYs.
  2. Lines 44-47: It would be very useful for readers to add how many decibels and above is considered harmful or excessive noise.
  3. Lines 56-64: Write clearly the objective of this review. In my opinion, "synthesising information" is an intrinsic aim of any review work. Normally in review papers that do NOT include meta-analysis, the verb used in the objective is to describe. But, taking into account that your work includes meta-analysis, you should use a more appropriate verb such as: estimate, evaluate, assess... "To assess the association between high noise annoyance and mental health disorders from the results of published articles" (or something like that, my native language is not English, I'm sure you will know how to express the aim of the study better, but I hope I have conveyed my suggestion correctly).
  4. What are “mental health outcomes”? Specify.
  5. Lines 56-64: A requirement of any systematic review is to include the research question and, if possible following the PICO or PICOS structure (see PRISMA checklists). Please, include a clear research question.

Methods

  1. Usually, the first paragraph is the “Study design”. Add if you consider it appropriate.
  2. The first line of methodology (“Study design”) should be the design of your work. Write clearly that it is a systematic review with meta-analysis. Next, you should state whether a protocol of this work has been published, whether it has not been published but you have one, and whether this work has been registered in PROSPERO or another platform for the same purpose. This is a very important point.
  3. In this first paragraph: Have you followed the standards of any guidelines for writing and structuring the review? PRISMA? If yes, please explain.
  4. Search strategy: why did you include "quality of life" and "perceived well-being" in your search strategy? I don't see any connection with your study objective.
  5. Inclusion criteria: It would be necessary to clarify that it is in the adult population (if it is).
  6. Inclusion criteria: In addition to the time limit on the publication of articles, have you taken into account the language in which they are written? Please clarify this and include it in the inclusion criteria if necessary.
  7. If a study did not include OR or RR, would you have included it for the "qualitative" analysis of your results or would you have discarded it? Please clarify this in the inclusion criteria, it is possible that this criterion is only applicable for the inclusion of an article in the meta-analysis.
  8. Definition of exposure: Please explain in detail what the scales or parameters HAN, HAV, HAVW consist of. When I try to look for information on ISO/TS 115 15666:2021, the content is fee-based. Have you bought these scales beforehand and filled them in based on the information from the studies you have found? This sub-section needs more information.
  9. Lines 91-94 and Lines 127-134: These paragraphs look like tasks that have been carried out after the screening and selection of articles has been completed. Please can you indicate whether a second screening of articles was carried out? In addition, it would be useful to add the date of the literature search in PubMed and Web of Science.
  10. Lines 141-142: Same comment as in the abstract.
  11. Risk of Bias: Explain in more detail the Methodology for Systematic Evidence Reviews for WHO Environmental Noise Guidelines for the European Region. How is it scored, is it validated, did you create it from an existing one?
  12. Statistical analysis: It is necessary to clarify how articles in which different mental disorders are studied at the same time have been treated in the data analysis (e.g. Beutel 2016, depression + anxiety).
  13. Statistical analysis: How did you score the responses to the bias scales for use in statistical analyses with R? Please, clarify.
  14. It would be advisable to make a sub-section on "Quality of evidence" and separate this information from "Statistical analysis". In addition, a more in-depth definition of the GRADE scale would be useful, especially the “overall” scores.

Results

  1. Figure 1: This flowchart is not the one indicated by PRISMA. You can find the template here: http://www.prisma-statement.org/PRISMAStatement/FlowDiagram Please change it and include the reasons for exclusion, at least in the abstract screening.
  2. Line 212: This is not entirely true. For example, in Beutel's study (2016), both depression and anxiety are studied. Could you clarify this sentence?
  3. Line 224: Include information from the R package in the statistical analysis section.
  4. Lines 227-229: It is clear to me that the main reason for the risk of bias is question D1 (lack of standardised definitions of high noise annoyance) but I do not understand why you indicate that the second reason is question D3. Questions D3, D4 and D5 have the same number of items with high risk of bias. Please, clarify.
  5. Lines 242-243: In my opinion, there is no need to warn readers about this since, if I have understood correctly, you have applied random effects, trying to deal with heterogeneity across studies. This is just a thought, you can keep the sentence if you want.
  6. Table A.5: Complete the table footer, adding the scale concerned and the definitions of the abbreviations listed in the table.
  7. Tables: Table A.4 and A.5 are the same reference [19]? Check.
  8. Tables: Table A.6: Complete the table footer, adding the definitions of the abbreviations listed in the table.

Discussion

  1. Limitations: The limitations you present are correct but few. It is necessary to mention the possible non-inclusion of relevant articles (limitation of all review articles), search strategy too broad and with non-MESH terms (which may increase the search results too much, making screening difficult), use of only two databases (limits the coverage of all possible articles), you have not used PsycINFO which is the most important mental health database (this is a very very important limitation), you have not assessed the methodological quality of the included articles (which may limit the reliability of your results), you have not established an age range of the participants (mental health disorders influence differently depending on age), most of the included articles are cross-sectional studies, which limits the establishment of causality in the association between mental disorder and high noise annoyance. All these aspects need to be explained in detail.
  2. If appropriate, you can include in the strengths the identification of a possible line of future research on the differences between men and women in the effect of high noise annoyance and depression and anxiety.

Conclusion

  1. Lines 412-413: This could be a strength rather than a conclusion.
  2. Lines 414-415: What link? Specify. Be direct in your response to your objective.

Thank you.

Round 2

Reviewer 2 Report

Thank you for your clear responses to my comments.